# A cluster-randomised controlled trial of the LifeLab education intervention to improve health literacy in adolescents

Kathryn Woods-Townsend[1,2]*, Polly Hardy-Johnson[3], Lisa Bagust[1], Mary Barker[2,3], Hannah Davey[1], Janice Griffiths[1,4], Marcus Grace[1], Wendy Lawrence[2,3], Donna Lovelock[1], Mark Hanson[2,5], Keith M. Godfrey[2,3,5‡], Hazel Inskip[2,3‡]

1 Southampton Education School, Faculty of Social Sciences, University of Southampton, Southampton, United Kingdom, 2 NIHR Southampton Biomedical Research Centre, University of Southampton and University Hospital Southampton NHS Foundation Trust, Southampton, United Kingdom, 3 MRC Lifecourse Epidemiology Unit, University of Southampton, Southampton, United Kingdom, 4 Mathematics and Science Learning Centre, Southampton Education School, Faculty of Social Sciences, University of Southampton, Southampton, United Kingdom, 5 Institute of Developmental Sciences, Faculty of Medicine, University of Southampton, Southampton, United Kingdom

‡ These authors are joint senior authors on this work.
* k.woods-townsend@soton.ac.uk

**Data Availability Statement:** All de-identified data used in the analysis presented in this manuscript

## Abstract

Adolescence offers a window of opportunity during which improvements in health behaviours could benefit long-term health, and enable preparation for parenthood—albeit a long way off, passing on good health prospects to future children. This study was carried out to evaluate whether an educational intervention, which engages adolescents in science, can improve their health literacy and behaviours. A cluster-randomised controlled trial of 38 secondary schools in England, UK was conducted. The intervention (LifeLab) drew on principles of education, psychology and public health to engage students with science for health literacy, focused on the message "Me, my health and my children's health". The programme comprised: • Professional development for teachers. • A 2–3 week module of work for 13-14-year-olds. • A "hands-on" practical health science day visit to a dedicated facility in a university teaching hospital. Data were collected from 2929 adolescents (aged 13–14 years) at baseline and 2487 (84.9%) at 12-month follow-up. The primary outcome was change in theoretical health literacy from pre- to 12 months post- intervention. This study is registered (ISRCTN71951436) and the trial status is complete. Participation in the LifeLab educational intervention was associated with an increase in the students' standardised total theoretical health literacy score (adjusted difference between groups = 0.27 SDs (95%CI = 0.12, 0.42)) at 12-month follow-up. There was an indication that intervention participants subsequently judged their own lifestyles more critically than controls, with fewer reporting their behaviours as healthy (53.4% vs. 59.5%; adjusted PRR = 0.94 [0.87, 1.01]). We conclude that experiencing LifeLab led to improved health literacy in adolescents and that they demonstrated a move towards a more critical judgement of health behaviour 12 months after the intervention. Further work is needed to examine whether this leads to sustained behaviour change, and whether other activities are needed to support this change.

are publicly available here: https://doi.org/10.5258/SOTON/D1606.

**Funding:** This work was supported by the British Heart Foundation (https://www.bhf.org.uk/) (PG/14/33/30827 and the National Institute for Health Research through the NIHR Southampton Biomedical Research Centre (https://www.uhs.nhs.uk/ClinicalResearchinSouthampton/Research/Facilities/NIHR-Southampton-Biomedical-Research-Centre/NIHRSouthamptonBiomedicalResearchCentre.aspx). KMG is supported by the UK Medical Research Council (https://mrc.ukri.org/) (MC_UU_12011/4), the National Institute for Health Research (as an NIHR Senior Investigator (https://www.nihr.ac.uk/) (NF-SI-0515-10042) and the European Union's Erasmus+ Capacity-Building ENeASEA Project and Seventh Framework Programme (https://ec.europa.eu/research/fp7/index_en.cfm) (FP7/2007-2013), projects EarlyNutrition and ODIN under grant agreement numbers 289346 and 613977. MAH was supported by the British Heart Foundation and HMI and MAH are supported by the European Union's Horizon 2020 research and innovation programme (733206, LifeCycle) (https://ec.europa.eu/programmes/horizon2020/en/tags/horizon-2020-research-and-innovation-programme). HD and LB are supported by Wessex Heartbeat (https://www.heartbeat.co.uk/) and HMI and MB are supported by the UK Medical Research Council. The funders had no role in the study design, data collection, data analysis, data interpretation or writing of the report.

**Competing interests:** The University of Southampton has received an unrestricted donation from Danone Nutricia to support LifeLab's work with schools and one of the authors (KMG) has received reimbursement for speaking at conferences sponsored by nutrition companies, and is part of an academic consortium that has received research funding from Abbott Nutrition, Nestec and Danone. This does not alter our adherence to PLOS ONE policies on sharing data and materials.

## Introduction

Adolescence is a critical developmental stage during which lifelong health behaviours can be established. At this time, targeted interventions may reduce the emergence or establishment of risk factors for health problems in adulthood, especially non-communicable diseases (NCDs) [1, 2]. In addition, a woman's diet and general health as she embarks on pregnancy can have profound and lasting effects on early development and on the lifelong health of her child [3]. The health behaviours of fathers are also important, both because they influence the health behaviours of their partners [4] and because the father's diet and lifestyle can have lasting biological effects on the offspring [5]. Thus, appreciating that adolescents are future parents provides a powerful argument for tailoring interventions to this age-group, but raises questions about the most appropriate forms of intervention to adopt for a population that can be hard to engage [6].

Transmission communication approaches (e.g. leaflets, advertising) designed to impart information are typically used in public health campaigns and, while these can raise awareness and may engender immediate behaviour change, sustaining these changes is challenging, particularly in adolescents [7]. Schools have long been seen as ideal settings for targeting public health interventions as they have the potential to reach a large population of children and young people across the socio-economic spectrum, and to provide a consistent and constant setting in which to engage them with peers and adults. Crucially, while external experts offer novelty and spark interest, students see teachers as trusted experts and who are also experienced in engaging with this age-group [8, 9]. No setting, other than the family, offers such a consistent, prolonged opportunity for engagement with children and young people [10–13] and specifically the opportunity to use education as a means to increase health literacy [14, 15].

Health literacy can be described as having the knowledge, skills, understanding and confidence to use health and care information and services and to apply these to lifestyle choices [16]. LifeLab is based on the premise that health and wellbeing are socio-scientific issues [17], that is, social issues with prominent scientific components[18, 19]. The LifeLab programme aims to foster an understanding of socio-scientific knowledge, alongside decision-making skills, thereby promoting the adolescents' sense of control over their lives and futures. It is suggested that one means of increasing health literacy is by developing scientific literacy; this is achieved by providing education for adolescents in 'science for health literacy' [20]. Research is emerging, that considers the effectiveness of such interventions on adolescents' behaviour [21]. Currently there are no validated instruments for assessing health literacy in this population. Consequently, during the development work for this trial, building on work carried out by Guttersrud et al. [22], but considering issues that were more appropriate for adolescents, a series of health literacy questions were identified. These have been used previously in feasibility and pilot studies [18, 23] and have been shown to be acceptable and understood by this age group, giving measurable differences over time.

The OECD PISA Science Framework defines scientific literacy as "the ability to engage with science-related issues, and with the ideas of science, as a reflective citizen" [24]. Knowledge of scientific principles, especially those related to human biology, critical thinking and the ability to make informed decisions about science-related issues, are attributes closely linked to health literacy. The obvious setting in which to teach scientific literacy is schools.

School-based interventions, aimed at impacting health outcomes, have, however, mostly been aimed at younger children rather than adolescents. Effects on health outcomes such as levels of obesity [25, 26], healthy eating [27], oral health [28], mental health [29] and physical activity [30] have been inconsistent, some showing positive effects, but some having no impact. Interventions targeted at adolescents (aged 10–19) [31] have also had limited success in

achieving their outcomes, possibly because these tend to adopt an instructional, transmission approach, simply providing information to students [32, 33]. Health behaviour change typically requires more than information [34].

Historically, there has been a lack of attention to explicit links to pedagogy and curriculum in the design of school-based health-related interventions [35]. An understanding of how young people learn and how best to set into students' current understanding both within a discipline, but importantly across the curriculum is key to delivering successful interventions within an educational setting. The LifeLab curriculum has been designed to be embedded within the school curriculum. It is an innovative, 'hands-on', educational intervention, which aims to promote health literacy through science engagement and increasing scientific literacy (see the LifeLab panel and TIDIER framework check list (S1 Checklist) for further detail) [23]. Based on research that is relevant to adolescents, LifeLab offers an opportunity to engage directly with the science behind the health messages. We argue that framing the science content within an educational intervention around 'developmental origins of health and disease' (DOHaD) concepts provides cutting-edge research stories that are engaging and relevant to adolescents. Exploiting the value of learning outside the classroom, which has been shown to produce learning experiences of long-lasting impact [28, 29], we established a purpose-built facility based in a large teaching hospital. As many young people have never been inside a hospital or a research laboratory, such an experience can make a great impression. LifeLab was also designed to maximise effectiveness by drawing on approaches shown to be successful and engaging but also to meet the needs of the teachers required to implement it. A systematic review has found that successful educational interventions are supported by inclusion of specifically tailored teacher professional development (Jacobs et al., under review). S1 Diagram presents a logic model which summarises the evidence, design, intended function and outcomes of LifeLab.

Pilot studies of the effect of LifeLab showed that participation in a science programme focusing on health, and experiencing learning within a hospital-based classroom, had a positive influence on adolescents' awareness of the importance of making healthy lifestyle choices [18, 23].

Here we present results from a cluster-randomised controlled trial (RCT) to evaluate whether taking part in LifeLab improved adolescents' nutrition and health literacy and whether participation changed how they viewed their own health behaviour 12 months after the experience.

## Materials and methods

### Study design

The cluster-RCT recruited adolescents aged 13–14 years from 38 state secondary schools/academies (approximately 2,500 participants in the South of England). Each school was randomly allocated to either 'control' or 'intervention' status. The study was approved by the Research Ethics Committee of the Faculty of Social Sciences, University of Southampton (ERG reference: 7892 amendment 1 (10/10/13), ERG reference: 12328 (14/11/14) and amendment 18817 (14/01/2016)). Written consent was obtained from participants and their parents. The study followed the CONSORT guidelines for the design and reporting of clinical trials (S2 CONSORT Checklist), and the CONSORT flow diagram for the study is shown in Fig 1. The full trial protocol has been published [41]. The concept for this project was informed by our pilot trial [23] and developed in 2014. In the initial phases, as this was an education RCT, the requirement to register as a clinical trial was not appreciated. Consequently, although the trial registration process was initiated in September 2014, final confirmation of registration did not

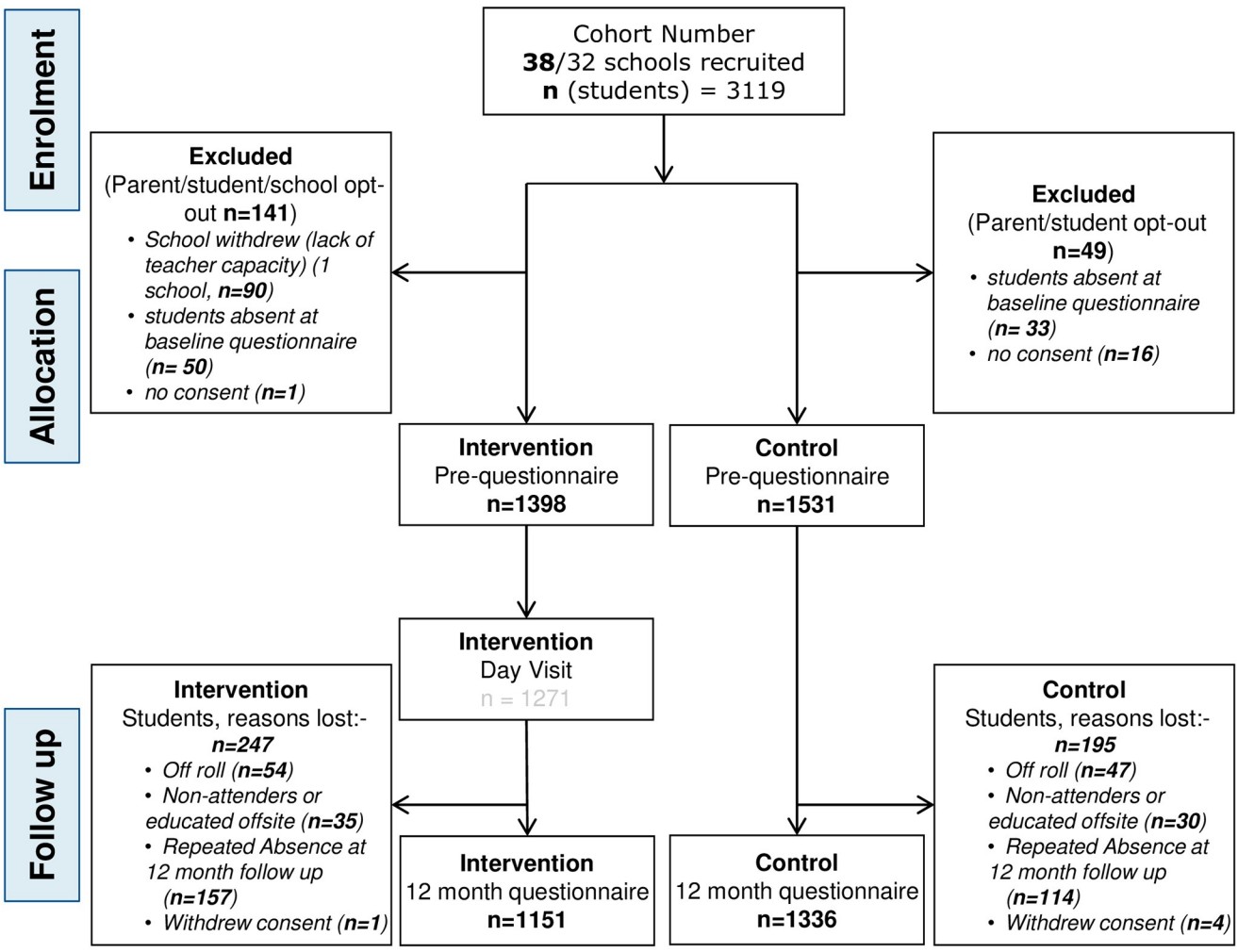

**Fig 1. CONSORT diagram to show participant flow through the trial.**

happen until March 2015. The trial is registered with International Standard Randomised Controlled Trials, number ISRCTN71951436, the authors confirm that all on-going and related trials for this intervention are registered.

## Participants

All state secondary schools within a two-hour travel radius of UHS were eligible. Though, we excluded independent, grammar and special schools. At an individual level, there were no exclusion criteria for students.

Participating schools were required to allocate a minimum of three 'middle ability' classes (~90 students). To minimise loss of participants to follow-up, student lists for each participating class were requested. Parent and student information sheets were provided for the schools to disseminate. Two versions of the information sheets were produced—'intervention' and 'control'.

For intervention schools, parental consent was opt-in. For the control schools, consent was opt-out at the request of the schools to place less of a burden on them. In all cases, student

assent was built into the questionnaire and, at any point, a student or parent could request that their information be withdrawn from the trial.

Recruitment of 38 schools was carried out between 1$^{st}$ July 2014 and 22$^{nd}$ October 2015. The first participant completed the baseline questionnaire in October 2014 and the last in July 2016. Follow-up took place between October 2015 and July 2017.

### Randomisation and masking

Schools were recruited via a number of routes: information flyers and targeted letters sent to Headteachers and Heads of Science; presentations at local network and leadership meetings; and word of mouth. In total 112 schools were made aware of the trial. We aimed to recruit the first 32 schools to respond but, over the time-period when recruitment was open, 38 responded and were therefore included. The distribution of schools which responded included a range of schools from across the South of England and hence were broadly representative of the general population. Of the schools recruited, 70% had been judged as Good or Outstanding at their most recent Ofsted inspection, slightly below the national average of 74%. Prior to baseline data collection, groups of recruited schools were randomised in blocks of even numbers to the control or intervention arms. The blocks were not randomly selected as we needed to accommodate the needs of the schools to be randomised quickly. We obtained the largest even number of schools we could in a short space of time and then randomised. The block sizes varied between 2 and 12. Once two schools had been recruited, the schools were numbered, documented and the date of return of Headteacher's agreement letter noted. Recruiting schools was an on-going process, the smallest block size being two and the largest twelve. Randomisation was conducted off-site to conceal allocation; researchers were unaware of the procedure, ensuring that prediction of allocation would not be possible. The randomisation process was conducted by a statistician at the MRC Lifecourse Epidemiology Unit with no knowledge of the schools in question, using computer-generated sequences. The nature of the intervention meant that it was not possible for researchers delivering the intervention, school staff, parents or students to be masked to group allocation.

### Procedures

In supervised class time at school, typically a science lesson, web-based questionnaires were administered to control and intervention participants at baseline and again approximately 12 months later. The LifeLab team supported teachers, from all schools during the process of administering the questionnaire; a member of the LifeLab team was present, but the students completed the questionnaires independently on their computers. During follow-up data collection, support was again provided as requested by the class teacher. The questionnaire took on average 20 minutes to complete. Students either completed the questionnaires on iPads brought into school as part of the data collection exercise, on school laptops or devices, or in a school computer room. Typically, classes consisted of 30 students, with the class teacher and a researcher present. Students with literacy issues were supported by the class teacher or researcher reading out the questions to them. A written script was read aloud to explain the process to the students, ensuring consistency across all control and intervention schools. Shortly after baseline data were collected, intervention schools began the programme, as set out in Box 1. Control schools received normal schooling, they were offered the chance to attend LifeLab in the subsequent year, with a different cohort of students.

## Box 1. LifeLab—Me, my health and my children's health

Based at University Hospital Southampton (UHS) NHS Foundation Trust, LifeLab is a state-of-the-art teaching laboratory dedicated to improving adolescent health through science engagement. Interventions targeted at adolescents have the potential for a "triple dividend" of benefits now, into future adult life and for the next generation of children [36]. This is mirrored in the LifeLab strapline: 'Me, my health and my children's health'. The LifeLab intervention aims to engage adolescents with the knowledge and understanding needed to enable them to make appropriate health choices—their health literacy—and to motivate them to change behaviour. This is delivered in an interactive and highly engaging format which sets scientific knowledge into a relevant and accessible context for this age group [21]. Students are able to learn how the nutrition of parents, children and adolescents influences health; to understand the impact of their lifestyle on their future risk of NCDs such as cardiovascular disease, type 2 diabetes, mental health and some types of cancer, and how their lifestyle impacts on the health of their future children.

LifeLab has three principal components:

**Component 1: Professional development for teachers**

A one-day training course at LifeLab explains its research basis, inspiring the teachers with cutting-edge science and the modules of student work, highlighting the opportunities for science enquiry skills to be incorporated in the classroom. Teachers receive training in Healthy Conversation Skills, to enable them to support their students to make behaviour changes and healthy choices [37].

**Component 2: A teaching module of work**

Linked to the National Curriculum for Science in England, an 8 lesson module for use with adolescents aged 13–14 years, encompasses pre- and post-LifeLab visit lessons, which engage students with authentic and relevant cutting edge research alongside opportunities to develop 'working scientifically' skills, to be delivered in school by teachers trained in Healthy Conversation Skills. The over-arching theme for the module of work looks at 'How scientists work'–how research questions are formulated, how studies are designed, how data are collected, analysed and communicated. The lessons are based on real-life research studies, such as the Southampton Women's Survey [38]. To prompt reflection and discussion among students, insights from behaviour change theory are applied to the development of the educational materials and the proposed learning activities. The Taxonomy of Behaviour Change Techniques is used to identify the behaviour change techniques underlying all learning activities [39].

**Component 3: A hands-on, practical health science day visit**

Held at LifeLab part-way through the module of work and delivered by LifeLab teachers, this visit gives students opportunities to experience a variety of ways to measure health, including studying carotid artery blood flow and structure using ultrasound, assessing body composition, performing lung function tests and measuring grip strength and hamstring flexibility. They also extract their own DNA and carry out gel electrophoresis experiments investigating the epigenetic processes which may mediate effects of early

development on later health. During this day, students have opportunities to participate in the 'Meet the Scientist' programme [40], which uses role models to break down misconceptions around what scientists do and what they are like.

## Outcomes

The primary outcome was a theoretical health literacy score. Following the approach devised by Guttersrud et al. [22], but guided by our previous pilot studies and PPI input from young people themselves, we identified a series of health literacy questions, based on those used in the pilot RCT [23], which assessed adolescents' knowledge of the way lifestyle choices can impact on health throughout the life course, and on the health of future generations. These were a series of five questions, four of which used a Likert scale, with five responses ranging from 'strongly disagree', (scored 1), to 'strongly agree' (scored 5), the fifth question "At what age do you think our nutrition starts to affect our future health?" has responses from before birth then in decades up to > 60 years. These were coded as >30 years = 1, 20–30 = 2, 10–20 = 3, 0–10 = 4, Before birth = 5. Responses to the individual health literacy questions were coded and totalled, and the scores ranged from 6 to 30, with 30 being the highest score possible. The baseline scores were then standardised to produce a normally distributed theoretical health literacy score with a mean of 0 and a standard deviation of 1, with higher scores indicating better health literacy. The follow-up scores were standardised to the mean and standard deviation of the baseline scores to allow change to be assessed. Secondary outcome measures were assessed with questions about students' reported health behaviours, and perceptions of the healthiness of their lifestyle, including understanding of cardiovascular disease and cancer risk. 'Prudent' diet score for each individual was an additional secondary outcome measure, generated using principal components analysis of data from the food frequency questionnaire on consumption of 15 specific food items. The second principal component gave positive weights to the foods and drinks that are generally considered healthy (e.g. fruit, vegetables, fish) and negative scores to those considered less healthy (e.g. biscuits, sweetened drinks, sausages). The score from this component at baseline was standardised as a prudent diet score in the same way as in previous studies [42]. The coefficients from this score were applied to the reported food and drinks consumption at follow-up (standardised to the mean and standard deviation of their values at baseline) to obtain a second prudent diet score for comparison with the baseline scores [42].

For the purposes of trial monitoring and reporting, an adverse event was considered to include any sign of distress related to diet, weight, or body image during the intervention or questionnaire completion, as reported by the school teachers, LifeLab teachers, researchers, parents or students themselves. None were reported.

## Statistical analysis

Using data (z-scores) from our previous research in young adults in and around Southampton [18], we estimated an intra-cluster correlation of 0.035 from previous data; to be conservative we rounded this up to 0.04. With three Year 9 (aged 13–14 years) classes from each school averaging 90 students per school and using 90 as our cluster size we estimated that 14 clusters in each group had 80% power at the 5% level of significance to detect a difference in change in primary outcome score of 0.25 SDs over a 12-month period between the intervention and control schools.

Descriptive statistics, including chi-squared tests and independent t-tests, were used to test differences at baseline between the control and intervention groups in school characteristics, participants' demographic characteristics (age, gender, and home deprivation score) and the primary outcome (theoretical health literacy score). Descriptive statistics were tested for differences between participants who dropped out and those who completed questionnaires at 12-month follow-up. To assess the level of deprivation of the area in which participants lived, the Income Deprivation Affecting Children Index (IDACI) [43] was obtained from home postcode, with higher scores indicating greater deprivation. An intention-to-treat analysis was performed using multi-level models to account for clustering [44]. Multiple linear regression analysis was used to compare theoretical health literacy and continuous secondary outcomes, in the intervention and control groups, adjusting for baseline values, gender and IDACI score. The categorical outcome variables were dichotomised and general linear mixed modelling was used to obtain prevalence rate ratios for the outcome in relation to the intervention. We used a Poisson model with robust standard errors and a log link function. Outcome variables were the results at the 12-month follow-up with adjustment for baseline responses and IDACI deprivation score included in all models. We also adjusted for gender since single-sex schools were included in the recruitment process and analysis showed small differences in gender distribution of the schools in the two arms of the trial. Planned subgroup analyses focused on whether there were different effects for boys and girls, and for more and less disadvantaged students (based on IDACI score). All models included schools as a random effect, assumed to be independently normally distributed. To assess the fit of the model, standardised residuals were plotted against fitted models and quantile-quantile (q-q) plots of the standardised residuals and the best linear unbiased predictors against the quantiles of the normal distribution were examined. All analyses were conducted in STATA.

## Results

Characteristics of the participating schools are presented in Table 1. Intervention schools had a slightly more disadvantaged profile: slightly lower attainment, and higher proportions of students on free school meals, with special educational needs, and with English as an additional language. Individual characteristics showed a mean age at recruitment of 13.9 years, with the intervention group living in more deprived areas and containing more girls than the control group. Adjustment for these factors was included in the analysis.

A total of 2929 participants (aged 13–14 years) completed the baseline data collection; small proportions of parents in both arms did not give consent for their child to participate (Intervention arm 0.06%; Control arm 1%). There was some loss to follow-up due to students moving schools during the school year (3.4% of students at 12mo follow-up), being educated off-site or non-attenders (2.2%), withdrawing consent (0.2%) and from repeated absence (9.3%), resulting in complete baseline and follow-up data from 2487 participants (85%). In the control group 12.7% (95%CI: 11.1 to 14.5%) of those completing the baseline questionnaire did not complete the follow-up one while the corresponding figures for the intervention were somewhat higher 17.7% (95%CI: 15.7 to 19.8%). Intervention schools had a more disadvantaged profile than control schools; as a consequence the number of intervention participants living in more deprived areas who dropped out of the study was also higher.

Adjusting for the *a priori* co-variates listed above, adolescents in the intervention group showed a greater improvement in the primary outcome measure, theoretical health literacy score, following the intervention than those in the control group by 0.27SDs (95%CI: 0.12, 0.42) (Table 2). The model appeared to be a reasonable fit with no apparent trend in the standardised residuals in relation to the fitted values and the q-q plots indicating that the

**Table 1. Baseline characteristics for intervention and control group.**

| | | Control Schools | Intervention Schools | Total | All eligible schools | National average |
|---|---|---|---|---|---|---|
| | | (n = 19) | (n = 19) | (n = 38) | | |
| **School Characteristics** Median (IQR) | 5 GCSEs A-C (%) | 60.0 | 52.0 | 54.5 | 55.5 | 57.3 |
| | | (48.0, 68.0) | (41.0, 65.0) | (47.0, 65.0) | | |
| | % of eligible pupils with special educational needs support | 13.0 | 13.7 | 13.7 | 12.8 | 11.4 |
| | | (7.1, 15.4) | (11.8, 16.3) | (10.7, 15.4) | | |
| | % of pupils with English not as first language | 2.4 | 3.5 | 2.7 | 8.0 | 14.7 |
| | | (1.8, 8.2) | (1.9, 10.3) | (1.8, 8.2) | | |
| | % of pupils eligible for free school meals at any time during the past 6 years | 20.0 | 25.1 | 24.8 | 25.35 | 29.0 |
| | | (13.4, 34.2) | (14.8, 40.6) | (14.2, 37.9) | | |
| | | Control | Intervention | Total | | |
| | | (n = 1531) | (n = 1398) | (n = 2929) | | |
| **Student Characteristics[#]** | Age (years), mean (SD) | 13.8 (0.38) | 14.0 (0.37) | 13.9 (0.38) | | |
| | Gender n (%) Girls | 711 (46.6) | 784 (56.2) | 1495 (51.2) | | |
| | Boys | 816 (53.4) | 611 (43.8) | 1427 (48.8) | | |
| | IDACI Score, median (IQR) | 0.11 | 0.15 | 0.12 | | |
| | | (0.06, 0.18) | (0.08, 0.29) | (0.07, 0.24) | | |

School-level characteristics and IDACI scores presented as medians (interquartile range (IQR)); Age presented as means (standard deviations (SD)). Gender is presented as n (%). 5 General Certificate Secondary Education (GCSE)s A-C = the percentage of pupils achieving 5+ A*-C including both English and Mathematics GCSEs.

[#]Student Characteristics data included from the n = 37 schools who participated—loss of 1 school post-recruitment, so no student data collected.

assumptions of normality were correct. The intraclass correlation coefficient for the model was 0.042, close to the value of 0.04 used in the original sample size calculation.

At the individual question level, adolescents in the intervention group were more likely to answer appropriately when asked 'At what age do you think our nutrition starts to affect our future health?' and increased their level of agreement with the statements: 'The food a woman eats when she is pregnant may affect the health of her baby when it is growing up'; 'The food I eat now may affect the health of any children I have in the future'; and 'The food a father eats before having a baby will affect the health of his children'. At baseline, adolescents in both groups showed a high level of agreement with the question "The food I eat now may affect my health in the future" but were less likely to understand the impact of parental diet (particularly father's diet) on the health of future children. The questions relating to impact of parental diet on future generations showed greater change for intervention participants.

When considering the secondary outcomes and comparing lifestyle perceptions, although there were no striking differences between intervention and control arms, the lowest PRR value related to a change in how adolescents perceived their own health behaviours (see Table 3); at follow-up, adolescents who had taken part in the intervention tended to judge their lifestyles to be less healthy than those adolescents in the control group. Specifically, intervention participants were less likely to report their lifestyle as 'very healthy' or 'healthy' (53.4%) compared with controls (59.5%) at the end of the study (PRR = 0.94; 95%CI = 0.87,

**Table 2. Change in the primary outcome of theoretical health literacy score.**

| Primary Outcome | | β (adjusted difference*) | 95%CI | Control Pre Mean (SD) | Control Post Mean (SD) | Intervention Pre Mean (SD) | Intervention Post Mean (SD) |
|---|---|---|---|---|---|---|---|
| Standardised Total Theoretical Health Literacy | SD units | 0.27 | (0.12, 0.42) | -0.03 (1.02) | 0.05 (1.00) | 0.04 (0.98) | 0.33 (1.06) |
| **Individual questions** | Outcome Response | PRR | 95%CI | Control Pre n (%) | Control Post n (%) | Intervention Pre N (%) | Intervention Post n (%) |
| At what age do you think our nutrition starts to affect our future health? | Before birth | 1.13 | (1.01, 1.27) | 564 (42.4) | 657 (49.4) | 475 (42.0) | 621 (54.9) |
| The food I eat now may affect my health in the future | Strongly agree or agree | 1.01 | (0.96, 1.07) | 1114 (83.6) | 1145 (85.9) | 920 (80.8) | 971 (85.3) |
| The food a woman eats when she is pregnant may affect the health of her baby when it is growing up | Strongly agree or agree | 1.23 | (1.13, 1.35) | 723 (54.3) | 691 (51.9) | 623 (54.8) | 713 (62.7) |
| The food I eat now may affect the health of any children I have in the future | Strongly agree or agree | 1.36 | (1.18, 1.56) | 449 (33.8) | 451 (33.9) | 419 (36.8) | 521 (45.8) |
| The food a father eats before having a baby will affect the health of his children | Strongly agree or agree | 1.68 | (1.33, 2.11) | 205 (15.4) | 208 (15.6) | 188 (16.5) | 295 (25.9) |

Generalised linear mixed models were used for the Theoretical Health Literacy score and changes in all responses analysed using generalised Poisson mixed models *adjusted for school clustering, baseline level of outcome measure, gender and baseline deprivation score. PRR = prevalence rate ratio. Pre and post values are only presented for students who had valid scores at both time points.

**Table 3. Changes in secondary outcomes.**

| Secondary Outcomes | | | β (adjusted difference*) | 95%CI | Control Pre Mean (SD) | Control Post Mean (SD) | Intervention Pre Mean (SD) | Intervention Post Mean (SD) |
|---|---|---|---|---|---|---|---|---|
| Students' understanding of their health behaviours | **Prudent diet score** | SD units | 0.02 | (-0.09, 0.13) | 0.11 (1.01) | -0.09 (0.94) | -0.11 (0.95) | -0.08 (0.92) |
| | Question | Outcome response | PRR | 95%CI | Control Pre n (%) | Control Post n (%) | Intervention Pre n (%) | Intervention Post n (%) |
| | **My lifestyle is usually** | Very healthy or healthy | 0.94 | (0.87, 1.01) | 776 (58.3) | 792 (59.5) | 632 (55.8) | 605 (53.4) |
| | **The food I eat is usually** | Very healthy or healthy | 0.96 | (0.86, 1.09) | 536 (40.4) | 546 (41.2) | 414 (36.7) | 430 (38.2) |
| | **How often do you do exercise which makes you out of breath?** | Once or more a day | 1.15 | (0.94, 1.41) | 306 (23.6) | 242 (18.7) | 239 (21.3) | 226 (20.1) |
| | **How often do you take long low level exercise (e.g. 20 minute walks, long swim)?** | Once or more a day | 0.97 | (0.86, 1.10) | 555 (42.9) | 668 (51.6) | 523 (46.6) | 574 (51.1) |
| Students' understanding of influences on their future risk of heart disease and lifestyle-related cancer | **There are certain things I can do to lower my risk of heart disease.** | Strongly agree or agree | 1.01 | (0.97, 1.04) | 1157 (89.6) | 1190 (92.1) | 961 (86.3) | 1019 (91.5) |
| | **There are certain things I can do to lower my risk of cancer.** | Strongly agree or agree | 1.06 | (0.99, 1.13) | 911 (70.7) | 979 (76.0) | 750 (67.8) | 874 (79.0) |

Changes in all responses were analysed using generalised Poisson mixed models *adjusted for school clustering, baseline level of outcome measure, gender and baseline deprivation score. PRR = prevalence rate ratio. Pre and post values are only presented for students who had valid scores at both time points.

1.01). However, there was no improvement in their prudent diet score (0.02SDs (95%CI: -0.08, 0.13)).

Among all participants at baseline, awareness that "there are certain things I can do to lower my risk of heart disease" was greater (88.1%) than that for "there are certain things I can do to lower my risk of cancer" (69.5%) (see Table 3).

Subgroup analysis showed no difference between the results for boys and girls, or for those living in more and less disadvantaged neighbourhoods.

## Discussion

Using a cluster RCT design and a large number of participants (2,487) with a high retention at follow-up at 12 months (85%), we have established that adolescents who are offered a science education intervention show increased health literacy manifesting in sustained change in knowledge of life-long effects of their health behaviours. These findings build on our previous work in which we showed, in a small pilot RCT of the LifeLab intervention, a change in adolescents' knowledge over a 12 month period [23]. In the present study we have shown change in the theoretical health literacy score as a result of participation in the LifeLab programme. The observed effect size was slightly larger than that used in the original power calculation, so this trial was not underpowered. Following participation, adolescents in the intervention arm tended to judge their lifestyles to be less healthy than before the intervention. During the intervention, adolescents learn about the impact of behaviour on health, which may have prompted critical reflection on their own health behaviour The COM-B (capability, opportunity and motivation) model of behaviour change proposes that capability encompasses knowledge, which is one precursor for changes in behaviour [45]. The experiential learning opportunities provided by the LifeLab programme, increased adolescents' knowledge and enabled them to access, understand and reflect on what they need to do to live healthier lives; this was demonstrated in their improved health literacy at follow-up. A recent systematic review which considered 17 studies, showed that increasing adolescent health literacy was linked to improvement of health behaviours [46] and commented on the paucity of research on adolescent health literacy and its impact on health behaviours. This suggests that improving health literacy may be connected to improving health outcomes. It has been suggested, that for public health interventions to be effective, they need to focus on upstream determinants of obesity, such as the environments in which people live; the physical, socio-cultural, economic and political contexts that influence diet and physical activity levels [25, 47, 48]. These upstream determinants, are largely beyond the control of the individual; thus an environment that is conducive to healthy choices is essential to support optimal individual choice. Our work suggests that education, specifically an individual's scientific and health literacy education, could be seen as a potential, but less conventional, upstream determinant which should be considered in the design and delivery of public health interventions. Further work is, however, essential to determine how to capitalise on engagement with science through programmes like LifeLab and build lasting changes in adolescent health behaviour.

Participants in both arms of the trial showed that their knowledge of the impact of lifestyle choices on prevention of cancer was markedly lower than that for cardiovascular disease. Recent evidence shows that the rate of obesity-related cancers is increasing more rapidly in young adults [49]. There is also a lack of awareness of the link between factors such as poor diet, sedentary behaviours and excess weight and risk of cancer [50]. As such, there is clearly a need to ensure that the correct messages are embedded in the education system [51], and importantly that there is a focus on improving the health literacy of adolescents.

## Comparison with other studies

The development of LifeLab drew on the experience of a similar laboratory-based facility for schools in Auckland, New Zealand, known as LENScience. Many children from across the country visited LENScience, and a before- and after- comparison was undertaken, albeit without a control group. Similar to our findings, the students' knowledge increased, but they also showed some limited impact on behaviour change [21]. However, conducting such studies at a time when adolescents are undergoing rapid changes in their behaviours means that the lack of a contemporaneous control group makes interpretation of such findings difficult. This was the rationale for the present study.

## Strengths and limitations

Historically, recruitment of schools has been a concern for educational RCTs [52, 53]. We have shown that providing opportunities linked to the National Curriculum, and which meet schools' needs, can provide a route to successful engagement; we over-recruited to this study. This is also a consequence of the project being embedded within the local education community and good working relationships being cultivated over many years. The overwhelming majority of parents supported the intervention, with more than 99% giving consent for their children to participate. The retention rate was high (85%), showing continued engagement from the schools in both the intervention and control arms.

Educational interventions delivered in school have the potential to interact with a large number of students, providing a means for public health interventions to have a wide reach. To increase the chance of effectiveness, however, an educational intervention needs to be delivered by those professionals who understand best how to engage with the appropriate age groups. Our intervention, which took into account the subject-specific constraints and other pressures under which teachers work, included as a key component the provision of face-to-face Healthy Conversation Skills training for teachers. This gave them communication skills to support students to reflect on their current health-related behaviours and plan changes to these. Behaviour change frameworks highlight the importance of a trusted 'facilitator' to deliver the intervention [54] and earlier research shows that school students prefer interventions delivered by teachers [8, 55]. Systematic reviews have shown that training for teachers prior to delivery of interventions is associated with effectiveness of the intervention [56] and Jacobs et al., under review.

A limitation of the study is that it was not possible to mask the schools, teachers, young people or intervention delivery staff to allocation, raising the risk of ascertainment bias. This is common in public health interventions. Standard protocols were followed for both the data collection and intervention delivery and a process evaluation was conducted alongside the intervention to determine both fidelity of implementation of the intervention and to document any indicators of bias. The RCT design was informed by discussion with school leaders, who deemed it necessary to have different consent procedures for the control and intervention arms. However, very few parents refused to give consent for their children to participate in either arm of the study, so we do not believe that this affected recruitment or led to differences between trial arms. Although we took a cluster randomisation approach, the relatively small numbers to include as individual clusters (n = 38) led to some imbalances, notably the follow-up was poorer in intervention schools which were in more deprived areas and this has to be considered a possible bias in interpreting the results. There were also more girls in the intervention arm. Our analysis was therefore adjusted for gender and deprivation level and took account of the baseline levels of outcome measures. There was a high retention rate for the follow-up measurements, though the participants lost to follow-up due to repeated absences at

school or educated off-site may represent the most vulnerable participants with the worst health behaviours; our analysis of those who were retained for follow-up suggested that those lost were more likely to be from areas of higher deprivation.

Finally, it was not possible to randomise the schools after baseline data collection, which would have been ideal for minimising risk of bias. Schools require timely information for planning their annual timetable and allocating curriculum time and the short time between baseline data collection and intervention delivery necessitated that intervention schools were made aware of their curriculum commitment as soon as possible.

Due to the paucity of data of adolescent health literacy and lack of validated measures for adolescent health literacy, we followed the approach devised by Guttersrud et al. [22], but considering issues that were more appropriate for adolescents, we identified a series of health literacy questions, used previously in feasibility and pilot studies [18, 23]. These have been shown to be acceptable and understood by this age group, and give measurable differences over time. Acknowledging this as a limitation of this current trial, but questioning what the gold standard against which such a scale could be validated, our on-going work aims to develop and validate an appropriate tool.

This intervention showed change in knowledge, which, in the outcome measures selected did not translate into behaviour change; this limitation of the current trial could reflect a need to include additional intervention elements which support participants to effect change in behaviour.

## Conclusion

We have shown that the LifeLab programme has the potential to engage adolescents with science, leading to sustained changes in health literacy and more critical judgement of their own behaviour. To increase effectiveness, educational interventions should be designed and implemented within the education system, not be imposed upon it, and be delivered by the education professionals, either subject or class teacher, or other suitably qualified educator, who has an understanding of the pedagogical approaches best employed to engage with the target age groups. This will also ensure that the intervention can be situated within the curriculum that the young people are following—reinforcing prior knowledge and showing the relevance to other aspects of their learning. There is growing evidence that multi-component behaviour change interventions are more effective than those using a single component [57] and an inter-disciplinary approach is essential for such interventions to have the best chance of success.

The LifeLab intervention, drawing on expertise from science education, psychology and public health has increased adolescent health literacy. Future work to capitalize on this and support changes in adolescent health behaviour may require the addition of other intervention elements. For example, digital technology may provide an opportunity for delivering tailored, responsive and engaging interventions in a form that may be particularly appealing to adolescents.

## Supporting information

**S1 Checklist. TIDIER framework checklist.**
(DOCX)

**S2 Checklist. CONSORT checklist.**
(DOC)

**S1 Diagram. Logic model for the LifeLab intervention.**
(TIF)

**S1 File. Ethics form.**
(DOCX)

**S2 File. Woods-Townsend et al. trials 2015.**
(PDF)

**S3 File.**
(DOC)

## Acknowledgments

We thank the participating students, their teachers and schools and the LifeLab staff for contributions to this project. Ken Cox provided computer support for the data management. Andri Christodoulou provided input to the design and development of the instruments to measure the primary and secondary outcomes.

## Author Contributions

**Conceptualization:** Kathryn Woods-Townsend, Janice Griffiths, Marcus Grace, Mark Hanson, Keith M. Godfrey, Hazel Inskip.

**Data curation:** Kathryn Woods-Townsend, Polly Hardy-Johnson, Lisa Bagust, Hannah Davey, Donna Lovelock, Hazel Inskip.

**Formal analysis:** Kathryn Woods-Townsend, Polly Hardy-Johnson, Hazel Inskip.

**Funding acquisition:** Janice Griffiths, Marcus Grace, Mark Hanson, Keith M. Godfrey, Hazel Inskip.

**Investigation:** Kathryn Woods-Townsend, Lisa Bagust, Hannah Davey, Wendy Lawrence, Donna Lovelock.

**Methodology:** Kathryn Woods-Townsend, Polly Hardy-Johnson, Mary Barker, Marcus Grace, Hazel Inskip.

**Project administration:** Kathryn Woods-Townsend, Lisa Bagust, Hannah Davey, Donna Lovelock, Hazel Inskip.

**Resources:** Kathryn Woods-Townsend, Lisa Bagust, Hannah Davey, Wendy Lawrence, Donna Lovelock.

**Supervision:** Kathryn Woods-Townsend, Mary Barker, Janice Griffiths, Marcus Grace, Wendy Lawrence, Mark Hanson, Keith M. Godfrey, Hazel Inskip.

**Validation:** Hazel Inskip.

**Visualization:** Kathryn Woods-Townsend, Polly Hardy-Johnson, Mary Barker, Mark Hanson, Keith M. Godfrey, Hazel Inskip.

**Writing – original draft:** Kathryn Woods-Townsend, Polly Hardy-Johnson, Keith M. Godfrey, Hazel Inskip.

**Writing – review & editing:** Kathryn Woods-Townsend, Polly Hardy-Johnson, Lisa Bagust, Mary Barker, Hannah Davey, Janice Griffiths, Marcus Grace, Wendy Lawrence, Donna Lovelock, Mark Hanson, Keith M. Godfrey, Hazel Inskip.

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
