## [Decision Letter · Decision Letter 0]

10 Jul 2020

PONE-D-19-24269

A cluster-randomised controlled trial of the LifeLab education intervention to improve health literacy in adolescents

PLOS ONE

Dear Dr. Woods-Townsend,

Thank you for submitting your manuscript to PLOS ONE. After careful consideration, we feel that it has merit but does not fully meet PLOS ONE’s publication criteria as it currently stands. Therefore, we invite you to submit a revised version of the manuscript that addresses the points raised during the review process. Please accept our apologies for the time it has taken to arrive at this decision.

The manuscript has been evaluated by four reviewers, and their comments are available below. Several of the reviewers have indicated the importance of the work and indicate that the manuscript will make a valuable contribution to the literature once published. However, the reviewers have also raised a number of concerns that need attention. They request additional information on methodological aspects of the study (such as the randomization procedure and the measures used in the study) and the statistical analysis performed and underlying assumptions. You will find their detailed comments below - please note that one reviewer has provided additional comments as an attachment to this letter.

Could you please revise the manuscript to carefully address the concerns raised?

We look forward to receiving your revised manuscript.

Kind regards,

George Vousden

Senior Editor

PLOS ONE

Journal Requirements:

2. Thank you for including your competing interests statement; "I have read the journal's policy and the authors of this manuscript have the following competing interests:

KMG has received reimbursement for speaking at conferences sponsored by nutrition companies, and is part of an academic consortium that has received research funding from Abbott Nutrition, Nestec and Danone. The University of Southampton has received an unrestricted donation from Danone Nutricia to support LifeLab’s work with schools."

3.

Thank you for submitting your clinical trial to PLOS ONE and for providing the name of the registry and the registration number. The information in the registry entry suggests that your trial was registered after patient recruitment began. PLOS ONE strongly encourages authors to register all trials before recruiting the first participant in a study.

1) your reasons for your delay in registering this study (after enrolment of participants started);

2) confirmation that all related trials are registered by stating: “The authors confirm that all ongoing and related trials for this drug/intervention are registered”.

Please also ensure you report the date at which the ethics committee approved the study as well as the complete date range for patient recruitment and follow-up in the Methods section of your manuscript.

Reviewers' comments:

Reviewer's Responses to Questions

**Comments to the Author**

1. Is the manuscript technically sound, and do the data support the conclusions?

Reviewer #1: Yes

Reviewer #2: Yes

Reviewer #3: Partly

Reviewer #4: Yes

2. Has the statistical analysis been performed appropriately and rigorously? 

Reviewer #1: Yes

Reviewer #2: Yes

Reviewer #3: Yes

Reviewer #4: Yes

3. Have the authors made all data underlying the findings in their manuscript fully available?

Reviewer #1: Yes

Reviewer #2: Yes

Reviewer #3: Yes

Reviewer #4: Yes

4. Is the manuscript presented in an intelligible fashion and written in standard English?

Reviewer #1: Yes

Reviewer #2: Yes

Reviewer #3: Yes

Reviewer #4: Yes

5. Review Comments to the Author

Reviewer #1: This is a model cluster randomized trial. There are a few statistical concerns that should be addressed in a revision:

1. The randomization procedure used appears to be a "random block design" according to the textbook (Rosenberger and Lachin, Randomization in Clinical Trials". It appears that blocks were randomly selected from size 2 to 12, yet there are only 38 schools, so it is not clear how many blocks were selected, and why the large number 12 was the upper limit. And were blocks sizes chosen uniformly among 2,4,6,8,10,12? Not clear at all.

2. The sample size is relatively small, justified by an intraclass correlation and measures of variability. Yet there is not mention in the discussion of whether the sample size assumptions were realized in the study? Was the sample size actually adequate? If not, what was the actual power for the primary outcome?

3. More information on the assumptions of the model, including the distributional assumption of the random effect, and appropriate residual analysis and diagnostics are needed, as would be required by any regression class.

4. In general, statistical methods for cluster randomized trials are very different from others, and appropriate statistical references should be made (e.g., Donner and Klar, Cluster Randomized Trial among many others).

Reviewer #2: General Comments

This is an interesting and useful article. By focussing on a non-clinical, adolescent population, it provides much-needed and useful evidence and authors should be commended on the reach and retention rate. Given the primary outcome is theoretical healthy literacy, readers need to be mindful that this is measured by the use of a non-validated (albeit soundly developed) questionnaire. This is clearly indicated by the authors and this paper should not be stalled by this. There was and is no such measurement tool yet available, and the questionnaire used was therefore appropriate. The large sample size is to be commended, both in terms of adolescent participants, and number of schools involved. The paper presents practical and feasible methods to improve health literacy in adolescents and is a welcome contribution to our knowledge in this area.

Overall comments

● The primary outcome was theoretical health literacy. Although line 395-402 details that no validated health literacy questionnaire for this age group currently exists, I think this should be more clearly signposted in the introduction to justify why the authors developed their own questionnaire. The use of feasibility and pilot studies to inform the questionnaire development should also be stressed earlier in the paper, to support the rigorous approach adopted by the authors.

● Methods/Results. Well written, and clear methodology and stat analysis sections. Statistics and results seem to be well presented also - though I say this with the caveat that my statistics knowledge is passable, but not expert. I would advocate however for including cross sectional baseline data (results) from the questionnaire - data that isn’t that readily available internationally. This should then be discussed and contextualised in the discussion section. If this data has been published elsewhere in it’s own right, then it can be referred to again here - but important for the reader to at least get a grasp of the findings cross sectionally for the population, and what this means in the context of international literature.

● Very short discussion offered, I would suggest that the Discussion be extended to more fully consider other relevant or similar programmes, and also to take into account the fact that there didn’t exist a validated questionnaire for HL of adolescents. IN addition, consideration of what was found cross sectionally for the entire cohort warrants presentation and then discussion here - probably at the opening of this Discussion section. What did you find out from these students? and what does it tell us about HL? The change that you detected, how well does it stand up to other studies? What was the magnitude of this change in terms of context? etc

Please see attached word document for tabulated line by line recommendations/changes required.

Reviewer #3: It’s very rare to see published RCTs of school-based health interventions, specifically cluster randomised trials, and as such, this article makes a valuable contribution to the literature. The choice of methodological approach, though challenging to implement, is appropriate, and the authors give a good justification of why blinding is not possible. Aside from the trial, documenting the innovative work of the ‘LifeLab’ programme is another good reason for publishing this article, and the groups desire to gather quality evidence of impact, emphasises their commitment to quality implementation. Having said this there are a few areas where the paper could be strengthened as discussed below:

1. Given previous LifeLab studies which shared evidence of the benefits of participation to students, an ethical question raised by the study is: what benefit did control schools gain from participating in the study? For example, were control schools included in the implementation in subsequent years? Or did they benefit in any other way, in order to balance the time taken for surveys? I appreciate that offering a different intervention for the controls, though potentially beneficial to the students, may compromise a clean comparison, however a sentence to justify this or how control schools benefitted from participation would be helpful.

2. Don’t the overlapping confidence intervals of baseline characteristics (disadvantage indicators) between control and intervention schools, suggest that the differences between arms are not statistically significant? Please clarify.

3. Overall participation/withdrawal/absence/declining data are inconsistently presented within the text of the findings section (lines 280-283). While ‘not consenting’ is given for both arms at baseline, the comparison between control and intervention for absence, withdrawal etc. has not been presented. Including these with their confidence intervals for the arms would allow the reader to judge statistical significance.

4. The repeated absence of 9.8% looks quite high. How does this compare to national figures? And is there a difference between intervention and control? (this is just a suggested point of discussion as opposed to a recommendation)

5. It’s noteworthy that responses to ‘The food I eat now may affect my health in the future’ did not change for both arms while statistically significant PRRs were observed for the other questions in table 2. At baseline, there was high agreement to this statement (The food I eat) for both arms, which suggests a good general background awareness of the impact of diet on individual health… but less so on the impact of parental diet on future generation health. The intervention appears to have had a modest and statistically significant impact on specific awareness in this area and is thus an important finding which could be highlighted in the discussion.

6. With regard to: ‘Specifically, intervention participants were less likely to report their lifestyle as ‘very healthy’ or ‘healthy’ (53.4%) compared with controls (59.5%) at the end of the study (PRR=0.94; 95%CI= 0.87, 1.01)’, that CI’s cross over 1.0 would suggest that though approaching significance, the finding is not statistically significant. This finding is also referred to in the abstract and drawn upon in the conclusion. While this finding is close to statistical significance, it may be more pertinent to reflect on: while the study has demonstrated impact on theoretical literacy, changing student health behaviours require corresponding contextual/societal shifts towards, for example, parental ability and desire and support change in diets and physical exercise, facilities where students can exercise, supportive social norms to changing behaviours etc. These are notoriously challenging to effect change, and are challenging to address through targeting schools alone. This does not in any way detract from the important value and impact of the intervention on theoretical literacy – perhaps the discussion could reflect on this?

7. The discussion reflects on the trial demonstrating a proof of concept that RCTs can successfully be done in schools – and for this, the team’s rigour in planning and implementing is to be commended. Also commendable is the participation rate of 85%, and it may be worthy of reflection: the team’s careful pre-trial engagement with the schools, and their flexibility in modifying consent processes to accommodate stakeholder views, is likely to have contributed to its success and high participation/retention rates.

Reviewer #4: This is an important study given the lack of research on health literacy in children. The study strengths include the large sample, novel intervention, and potential impact on school curriculum focus on health literacy. However, key methodological details have been omitted.

Abstract

Page 3: Line 53, “enable preparation for parenthood” might be too large of a stretch based on the results. Can be re-worded to say “and possibly passing on good health prospects to future children.” This point also relates to the 1st paragraph on page 4. Developmentally, adolescents are more concerned with immediate gratification and preparing for parenthood is far-fetched for them.

Introduction

Page 4: Are there any studies showing how scientific literacy or science for health literacy is linked to adolescents’ behaviors? If so, please include

Page 5: Expand on “inconsistent findings” in line 117 and “limited success” in line 119.

Page 5: Lines 116 to 121, it is unclear whether the authors are referring to health behavior interventions in general or scientific literacy interventions specifically.

Page 7: Figure 1, under component 2, a sample outline of one of the lessons in the module would be help readers visualize the intervention.

Method

Page 9: Given that health literacy is the primary outcome. More information on the measure is warranted (e.g., how many items, response options, measurement statistics, how was the measure developed and validated in the pilot RCT, how does the measure map on to the intervention objectives) in this section. The reader should not need to scroll to the results table to figure out number of items and response options.

- Please report on the overall mean and std deviation of the total health literacy score and the individual items.

Page 9: Line 201, what kind of support was provided to teachers? Please provide examples and specify if this support was equal across sites.

Results

Page 11: Please include other relevant student characteristics (e.g., race and/or ethnicity) to corroborate and illustrate what the authors mean by “broadly representative of the general population” in line 185.

Page 12: Completion statistics for intervention is missing – how many did not complete, average dose of intervention, outliers with school with very low/high dosage and how were these characteristics adjusted for in the analyses? Intervention fidelity – what were the results of the process evaluation?

Page 12: Lines 284-285, were these ‘small differences’ statistically significant? Please provide percentages to be consistent with how information is presented in the rest of the paragraph.

Discussion

Page 15: Lines 340-341, define upstream determinants and expound on how health literacy can be conceptualized and treated as an upstream determinant in the context of this study.

6. PLOS authors have the option to publish the peer review history of their article (what does this mean?). If published, this will include your full peer review and any attached files.

Reviewer #1: No

Reviewer #2: No

Reviewer #3: No

Reviewer #4: No

---

## [Author Response · Author response to Decision Letter 0]

27 Oct 2020

Response to Reviewers' comments document is included in attachments

---

## [Decision Letter · Decision Letter 1]

12 Apr 2021

A cluster-randomised controlled trial of the LifeLab education intervention to improve health literacy in adolescents

PONE-D-19-24269R1

Dear Dr. Woods-Townsend,

We’re pleased to inform you that your manuscript has been judged scientifically suitable for publication and will be formally accepted for publication once it meets all outstanding technical requirements.

Kind regards,

Christopher M Doran, BEc (Hons) PhD

Academic Editor

PLOS ONE

Additional Editor Comments (optional):

Reviewers' comments:

Reviewer's Responses to Questions

**Comments to the Author**

1. If the authors have adequately addressed your comments raised in a previous round of review and you feel that this manuscript is now acceptable for publication, you may indicate that here to bypass the “Comments to the Author” section, enter your conflict of interest statement in the “Confidential to Editor” section, and submit your "Accept" recommendation.

Reviewer #1: All comments have been addressed

Reviewer #2: All comments have been addressed

Reviewer #3: All comments have been addressed

2. Is the manuscript technically sound, and do the data support the conclusions?

Reviewer #1: Yes

Reviewer #2: Yes

Reviewer #3: Yes

3. Has the statistical analysis been performed appropriately and rigorously? 

Reviewer #1: (No Response)

Reviewer #2: Yes

Reviewer #3: Yes

4. Have the authors made all data underlying the findings in their manuscript fully available?

Reviewer #1: (No Response)

Reviewer #2: Yes

Reviewer #3: Yes

5. Is the manuscript presented in an intelligible fashion and written in standard English?

Reviewer #1: (No Response)

Reviewer #2: Yes

Reviewer #3: Yes

6. Review Comments to the Author

Reviewer #1: My comment did not indicate that you should recompute your sample size, only to verify that the assumptions you used in the design actually were reasonable. So, no, I am not asking you to bias the study. Please add your following sentence to the conclusions in the appropriate spot:

We note that "the observed effect size was slightly larger than that used in the original power calculation,

so this trial was not underpowered."

Reviewer #2: This manuscript reads well and I am happy to all requested edits have been made. It will make a great contribution to the field.

Reviewer #3: Many thanks for responding to all the comments, they have been adequately addressed. I feel that the manuscript is worthy of publication.

7. PLOS authors have the option to publish the peer review history of their article (what does this mean?). If published, this will include your full peer review and any attached files.

Reviewer #1: No

Reviewer #2: No

Reviewer #3: No

---

## [Editor Report · Acceptance letter]

23 Apr 2021

PONE-D-19-24269R1 

A cluster-randomised controlled trial of the LifeLab education intervention to improve health literacy in adolescents 

Dear Dr. Woods-Townsend:

I'm pleased to inform you that your manuscript has been deemed suitable for publication in PLOS ONE. Congratulations! Your manuscript is now with our production department. 

Kind regards, 

on behalf of

Professor Christopher M Doran 

Academic Editor

PLOS ONE